# AUTOREGRESSIVE CONVOLUTIONAL NEURAL NETWORKS FOR ASYNCHRONOUS TIME SERIES

## ABSTRACT

We propose *Significance-Offset Convolutional Neural Network*, a deep convolutional network architecture for regression of multivariate asynchronous time series. The model is inspired by standard autoregressive (AR) models and gating mechanisms used in recurrent neural networks. It involves an AR-like weighting system, where the final predictor is obtained as a weighted sum of adjusted regressors, while the weights are data-dependent functions learnt through a convolutional network. The architecture was designed for applications on asynchronous time series and is evaluated on such datasets: a hedge fund proprietary dataset of over 2 million quotes for a credit derivative index, an artificially generated noisy autoregressive series and household electricity consumption dataset. The proposed architecture achieves promising results as compared to convolutional and recurrent neural networks. The code for the numerical experiments and the architecture implementation will be shared online to make the research reproducible.

## 1 INTRODUCTION

Time series forecasting is focused on modeling the predictors of future values of time series given their past. As in many cases the relationship between past and future observations is not deterministic, this amounts to expressing the conditional probability distribution as a function of the past observations:

$$p(X_{t+d}|X_t, X_{t-1}, \ldots) = f(X_t, X_{t-1}, \ldots). \tag{1}$$

This forecasting problem has been approached almost independently by econometrics and machine learning communities.

In this paper we examine the capabilities of convolutional neural networks (CNNs), (Lecun et al., 1998) in modeling the conditional mean of the distribution of future observations; in other words, the problem of *autoregression*. We focus on time series with multivariate and noisy signal. In particular, we work with financial data which has received limited *public* attention from the deep learning community and for which nonparametric methods are not commonly applied. Financial time series are particularly challenging to predict due to their low signal-to-noise ratio (cf. applications of Random Matrix Theory in econophysics (Laloux et al., 2000; Bun et al., 2017)) and heavy-tailed distributions (Cont, 2001). Moreover, the predictability of financial market returns remains an open problem and is discussed in many publications (cf. efficient market hypothesis of Fama (1970)).

A common situation with financial data is that the same signal (e.g. value of an asset) is observed from different *sources* (e.g. financial news, analysts, portfolio managers in hedge funds, market-makers in investment banks) in asynchronous moments of time. Each of these sources may have a different bias and noise with respect to the original signal that needs to be recovered (cf. time series in Figure 1). Moreover, these sources are usually strongly correlated and lead-lag relationships are possible (e.g. a market-maker with more clients can update its view more frequently and precisely than one with fewer clients). Therefore, the significance of each of the available past observations might be dependent on some other factors that can change in time. Hence, the traditional econometric models such as AR, VAR, VARMA (Hamilton, 1994) might not be sufficient. Yet their relatively good performance motivates coupling such linear models with deep neural networks that are capable of learning highly nonlinear relationships.

---

[1]iTraxx Europe Main Index, a tradable Credit Default Swap index of 125 investment grade rated European entities.

For these reasons, we propose *Significance-Offset Convolutional Neural Network*, a Convolutional Network extension of standard autoregressive models (Sims, 1972; 1980) equipped with a nonlinear weighting mechanism and provide empirical evidence on its competitiveness with standard multilayer CNN and recurrent Long-Short Term Memory network (Hochreiter & Schmidhuber, 1997). The mechanism is inspired by the gating systems that proved successful in recurrent neural networks (Hochreiter & Schmidhuber, 1997; Chung et al., 2014) and highway networks (Srivastava et al., 2015).

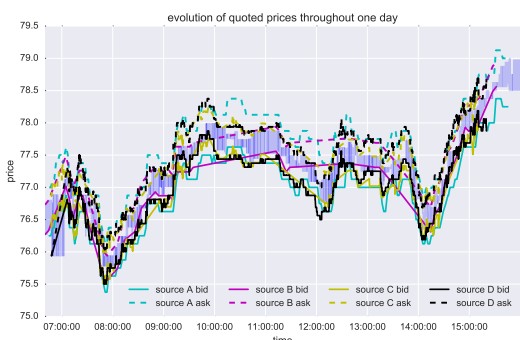

Figure 1: Quotes from four different market participants (sources) for the same CDS[1] throughout one day. Each trader displays from time to time the prices for which he offers to buy (*bid*) and sell (*ask*) the underlying CDS. The filled area marks the difference between the best sell and buy offers (*spread*) at each time.

## 2 RELATED WORK

### 2.1 TIME SERIES FORECASTING

Literature in time series forecasting is rich and has a long history in the field of econometrics which makes extensive use of *linear* stochastic models such as AR, ARIMA and GARCH processes to mention a few. Unlike in machine learning, research in econometrics is more focused on explaining variables rather than improving out-of-sample prediction power. In practice, one can notice that these models 'over-fit' on financial time series: their parameters are unstable and out-of-sample performance is poor.

Reading through recent proceedings of the main machine learning venues (e.g. ICML, NIPS, AISTATS, UAI), one can notice that time series are often forecast using Gaussian processes (Petelin et al., 2011; Tobar et al., 2015; Hwang et al., 2016), especially when time series are irregularly sampled (Cunningham et al., 2012; Li & Marlin, 2016). Though still largely independent, researchers have started to "bring together the machine learning and econometrics communities" by building on top of their respective fundamental models yielding to, for example, the Gaussian Copula Process Volatility model (Wilson & Ghahramani, 2010). Our paper is in line with this emerging trend by coupling AR models and neural networks.

Over the past 5 years, deep neural networks have surpassed results from most of the existing literature in many fields (Schmidhuber, 2015): computer vision (Krizhevsky et al., 2012), audio signal processing and speech recognition (Sak et al., 2014), natural language processing (NLP) (Bengio et al., 2003; Collobert & Weston, 2008; Grave et al., 2016; Jozefowicz et al., 2016). Although sequence modeling in NLP, i.e. prediction of the next character or word, is related to our forecasting problem (1), the nature of the sequences is too dissimilar to allow using the same cost functions and architectures. Same applies to the adversarial training proposed by Mathieu et al. (2016) for video frame prediciton, as such approach favors *most plausible* scenarios rather than outputs *close* to all possible outputs, while the latter is usually required in financial time series due to stochasticity of the considered processes.

Literature on deep learning for time series forecasting is still scarce (cf. Gamboa (2017) for a recent review). Literature on deep learning for *financial* time series forecasting is even scarcer though interest in using neural networks for financial predictions is not new (Mozer, 1993; McNelis, 2005). More recent papers include Sirignano (2016) that used 4-layer perceptrons in modeling price change distributions in Limit Order Books, and Borovykh et al. (2017) who applied more recent WaveNet architecture of van den Oord et al. (2016a) to several short univariate and bivariate time-series (including financial ones). Despite claim of applying deep learning, Heaton et al. (2016) use autoencoders with a single hidden layer to compress multivariate financial data. Besides these and claims of secretive hedge funds (it can be marketing surfing on the deep learning hype), no promising results or innovative architectures were publicly published so far, to the best of our knowledge. In this paper, we investigate the gold standard architectures' (simple Convolutional Neural Network (CNN), Residual Network, multi-layer LSTM) capabilities on AR-like artificial asynchronous and noisy time series, and on real financial data from the credit default swap market where some inefficiencies may exist, i.e. time series may not be totally random.

## 2.2 GATING AND WEIGHTING MECHANISMS

Gating mechanisms for neural networks were first proposed by Hochreiter & Schmidhuber (1997) and proved essential in training recurrent architectures (Jozefowicz et al., 2016) due to their ability to overcome the problem of vanishing gradient. In general, they can be expressed as

$$f(x) = c(x) \otimes \sigma(x), \tag{2}$$

where $f$ is the output function, $c$ is a 'candidate output' (usually a nonlinear function of $x$), $\otimes$ is an element-wise matrix product and $\sigma : \mathbb{R} \rightarrow [0, 1]$ is a sigmoid nonlinearity that controls the amount of the output passed to the next layer (or to further operations within a layer). Appropriate compositions of functions of type 2 lead to the popular recurrent architectures such as LSTM (Hochreiter & Schmidhuber, 1997) and GRU (Chung et al., 2014).

A similar idea was recently used in construction of highway networks (Srivastava et al., 2015) which enabled successful training of deeper architectures. van den Oord et al. (2016b) and Dauphin et al. (2016) proposed gating mechanisms (respectively with hyperbolic tangent and linear 'candidate outputs') for training deep convolutional neural networks.

The gating system that we propose is aimed at weighting a number of different 'candidate predictors' and therefore is most closely related to the *softmax* gating used in MuFuRU (Multi-Function Recurrent Unit, Weissenborn & Rocktäschel (2016)), i.e.

$$f(x) = \sum_{l=1}^{L} p^l(x) \otimes f^l(x), \quad p(x) = \text{softmax}(\widehat{p}(x)), \tag{3}$$

where $(f^l)_{l=1}^{L}$ are candidate outputs (*composition operators* in MuFuRu) and $(\widehat{p}^l)_{l=1}^{L}$ are linear functions of the inputs.

The idea of weighting outputs of the intermediate layers within a neural networks is also used in attention networks (See e.g. Cho et al. (2015)) that proved successful in such tasks as image captioning and machine translation. Our approach is similar as the separate inputs (time series steps) are weighted in accordance with learned functions of these inputs, yet different since we model these functions as multi-layer CNNs (instead of projections on learned directions) and since we do not use recurrent layers. The latter is important in the above mentioned tasks as it enables the network to remember the parts of the sentence/image already translated/described.

## 3 MOTIVATION

Time series observed in irregular moments of time make up significant challenges for learning algorithms. Gaussian processes provide useful theoretical framework capable of handling asynchronous data; however, due to assumed Gaussianity they are inappropriate for financial datasets, which often follow fat-tailed distributions (Cont, 2001). On the other hand, prediction of even a simple autoregressive time series such us AR(2) given by $X(t) = \alpha X(t-1) + \beta X(t-2) + \varepsilon(t)$ [2] may involve highly nonlinear functions when sampled irregularly. Precisely, it can be shown that the conditional expectation

$$\mathbb{E}[X(t)|X(t-1), X(t-k), k] = a_k X(t-1) + b_k X(t-k), \tag{4}$$

where $a_k$ and $b_k$ are rational functions of $\alpha$ and $\beta$ (See Appendix A for the proof). This would not be a problem if $k$ was fixed, as then one would be interested in estimating of $a_k$ and $b_k$ directly; this, however, is not the case with asynchronous sampling. When $X$ is an autoregressive series of higher order and more past observations are available, the analogous expectation $\mathbb{E}[X(t_n)|\{X(t_{n-m}), m = 1, \ldots, M\}]$ would involve more complicated functions that in general may not possess closed forms.

In real–world applications we often deal with multivariate time series whose dimensions are observed separately and asynchronously. This adds even more difficulty to assigning appropriate weights to the past values, even if the underlying data structure is linear. Furthermore, appropriate representation of such series might be not obvious as aligning such series at fixed frequency may lead to loss of information (if too low frequency is chosen) or prohibitive enlargement of the dataset (especially when durations have varying magnitudes), see Figure 2A. As an alternative, we might

---

[2]Where $\varepsilon(t)$ is an error term independent of $\{X(s) : s < t\}$.

consider representing separate dimensions as a single one with dimension and duration indicators as additional features. Figure 2B presents this approach, which is going to be at the core of the proposed architecture.

A natural model for prediction of such series could be an LSTM, which, given consecutive input values and respective durations $(X(t_n), t_n - t_{n-1}) =: x_n$ in each step would memorize the series values and weight them at the output according to the durations. However, the drawback of such approach lies in imbalance between the needs for memory and for nonlinearity: the weights that such network needs to assign to the memorized observations potentially require several layers of nonlinearity to be computed properly, while past observations might just need to be memorized as they are.

For these reasons we shall consider a model that combines simple autoregressive approach with neural network in order to allow learning meaningful data-dependent weights

$$\mathbb{E}[x_n | \{x_{n-m}, m = 1, \ldots, M\}] =$$

$$= \sum_{m=1}^{M} \alpha_m(x_{n-m}) \cdot x_{n-m} \quad (5)$$

where $(\alpha_m)_{m=1}^{M}$ satisfying $\alpha_1 + \cdots + \alpha_M \leq 1$ are modeled using neural network. To allow more flexibility and cover situations when e.g. observed values of $x$ are biased, we should consider the summation over terms $\alpha_m(x_{n-m}) \cdot f(x_{n-m})$, where $f$ is also a neural network. We formalize this idea in Section 4.

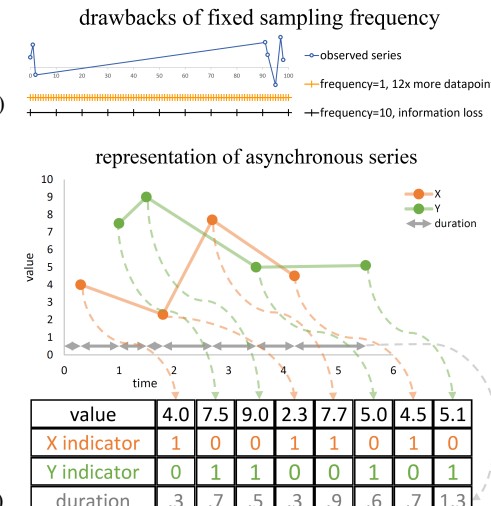

Figure 2: (A) Fixed sampling frequency and it's drawbacks; keeping all available information leads to much more datapoints. (B) Proposed data representation for the asynchronous series. Consecutive observations are stored together as a single *value* series, regardless of which series they belong to; this information, however, is stored in *indicator* features, alongside durations between observations.

| value | 4.0 | 7.5 | 9.0 | 2.3 | 7.7 | 5.0 | 4.5 | 5.1 |
|---|---|---|---|---|---|---|---|---|
| X indicator | 1 | 0 | 0 | 1 | 1 | 0 | 1 | 0 |
| Y indicator | 0 | 1 | 1 | 0 | 0 | 1 | 0 | 1 |
| duration | .3 | .7 | .5 | .3 | .9 | .6 | .7 | 1.3 |

## 4 MODEL ARCHITECTURE

Suppose that we are given a multivariate time series $(x_n)_n \subset \mathbb{R}^d$ and we aim to predict the conditional future values of a subset of elements of $x_n$

$$y_n = \mathbb{E}[x_n^I | \{x_{n-m}, m = 1, 2, \ldots\}], \quad (6)$$

where $I = \{i_1, i_2, \ldots i_{d_I}\} \subset \{1, 2, \ldots, d\}$ is a subset of features of $x_n$. Let $\mathbf{x}_n^{-M} = (x_{n-m})_{m=1}^{M}$. We consider the following estimator of $y_n$

$$\hat{y}_n^{(i)} = \sum_{m=1}^{M} \left[ F(\mathbf{x}_n^{-M}) \otimes \boldsymbol{\sigma}(S(\mathbf{x}_n^{-M})) \right]_{im}, i \in 1, 2, \ldots, d_I, \quad (7)$$

where

- $F, S : \mathbb{R}^{d \times M} \to \mathbb{R}^{d_I \times M}$ are neural networks described below,
- $\boldsymbol{\sigma}$ is a normalized activation function independent on each row, i.e.

$$\boldsymbol{\sigma}((a_1^T, \ldots, a_{d_I}^T)^T) = (\sigma(a_1)^T, \ldots, \sigma(a_{d_I})^T)^T \quad (8)$$

  for any $a_1, \ldots, a_{d_I} \in \mathbb{R}^M$ and $\sigma$ such that $\sigma(a)^T \mathbf{1}_M = 1$ for any $a \in \mathbb{R}^M$.

- $\otimes$ is Hadamard (element-wise) matrix multiplication.

The summation in 7 goes over the columns of the matrix in bracket; hence the $i$-th element of the output vector $\hat{y}_n$ is a linear combination of the $i$-th row of the matrix $F(\mathbf{x}_n^{-M})$. We are going to consider $S$ to be a fully convolutional network (composed solely of convolutional layers) and $F$ of the form

$$F(\mathbf{x}_n^{-M}) = W \otimes \left[\text{off}(x_{n-m}) + x_{n-m}^I\right]_{m=1}^M \tag{9}$$

where $W \in \mathbb{R}^{d_I \times M}$ and off : $\mathbb{R}^d \to \mathbb{R}^{d_I}$ is a multilayer perceptron. In that case $F$ can be seen as a sum of projection ($\mathbf{x} \mapsto \mathbf{x}^I$) and a convolutional network with all kernels of length 1. Equation (7) can be rewritten as

$$\hat{y}_n = \sum_{m=1}^M W_m \otimes (\text{off}(x_{n-m}) + x_{n-m}^I) \otimes \sigma(S_m(\mathbf{x}_n^{-M})), \tag{10}$$

where $W_m, S_m(\cdot)$ are $m$-th columns of matrices $W$ and $S(\cdot)$.

We will call the proposed network a *Significance-Offset Convolutional Neural Network* (SOCNN), while off and $S$ respectively the *offset* and *significance* (sub)networks. The network scheme is shown in Figure 3. Note that when off $\equiv 0$ and $\sigma \equiv 1$ the model simplifies to the collection of $d_I$ separate $AR(M)$ models for each dimension.

**Interpretation of the components**
Note that the form of Equation (10) enforces the separation of temporal dependence (obtained in weights $W_m$), the local significance of observations $S_m$ ($S$ as a convolutional network is determined by its filters which capture local dependencies and are independent on the relative position in time) and the predictors off($x_{n-m}$) that are completely independent on position in time. This provides some amount of interpretability of the fitted functions and weights. For instance, each of the past observations provides an adjusted single regressor for the target variable through the offset network. Note that due to asynchronous sampling procedure, consecutive values of $x$ might be heterogenous, hence On the other hand, significance network provides data-dependent weights for all regressors and sums them up in autoregressive manner. Figure 7 in Appendix E.2 shows sample significance and offset activations for the trained network.

**Relation to asynchronous data**
As mentioned before, one of the common problems with time series are the varying durations between consecutive observations. A simple approach at data-preprocessing level is aligning the observations at some chosen frequency by e.g. duplicating or interpolating observations. This, however, might extend the size of an input and, therefore, model complexity.

Figure 3: A scheme of the proposed SOCNN architecture. The network preserves the time-dimension up to the top layer, while the number of features per timestep (filters) in the hidden layers is custom. The last convolutional layer, however, has the number of filters equal to dimension of the output. The *Weighting* frame shows how outputs from offset and significance networks are combined in accordance with Eq. 10.

The other idea is to treat the duration and/or time of the observation as another feature, as presented in Figure 2B. This approach is at the core of the SOCNN architecture: the significance network is aimed at learning the high-level features that indicate the relative importance of past observations, which, as shown in Section 3, could be predominantly dependent on time and duration between observations.

**Loss function**

$L^2$ error is a natural loss function for the estimators of expected value

$$L^2(y, y') = \|y - y'\|^2. \tag{11}$$

As mentioned above, the output of the offset network can be seen as a collection of separate predictors of the changes between corresponding observations $x_{n-m}^I$ and the target variable $y_n$

$$\text{off}(x_{n-m}) \simeq y_n - x_{n-m}^I. \tag{12}$$

For that reason, we consider the *auxiliary loss* function equal to mean squared error of such intermediate predictions

$$L^{aux}(\mathbf{x}_n^{-M}, y_n) =$$

$$\frac{1}{M} \sum_{m=1}^{M} \|\text{off}(x_{n-m}) + x_{n-m}^I - y_n\|^2. \tag{13}$$

The total loss for the sample $(\mathbf{x}_n^{-M}, y_n)$ is therefore given by

$$L^{tot}(\mathbf{x}_n^{-M}, y_n) = L^2(\widehat{y}_n, y_n) + \alpha L^{aux}(\mathbf{x}_n^{-M}, y_n), \tag{14}$$

where $\widehat{y}_n$ is given by Eq. 10 and $\alpha \geq 0$ is a constant. In Section 5.2 we discuss the empirical findings on the impact of positive values of $\alpha$ on the model training and performance, as compared to $\alpha = 0$ (lack of auxiliary loss).

## 5 EXPERIMENTS

We evaluate the proposed model on a financial dataset of bid/ask quotes sent by several market participants active in the credit derivatives market, artificially generated datasets and household electric power consumption dataset available from UCI repository (Lichman, 2013), comparing its performance with simple CNN, single- and multi-layer LSTM (Hochreiter & Schmidhuber, 1997) and 25-layer ResNet (He et al., 2015).

Apart from performance evaluation of SOCNNs, we discuss the impact of the network components, such as auxiliary loss and the depth of the offset sub-network.

The details of the training process and hyperparameters used in the proposed architecture as well as in benchmark models are presented in C.

### 5.1 DATASETS

**Artificial data**

We test our network architecture on the artificially generated datasets of multivariate time series. We consider two types of series:

1. *Synchronous series.* The series of $K$ noisy copies ('sources') of the same univariate autoregressive series ('base series'), observed together at random times. The noise of each copy is of different type.

2. *Asynchronous series.* The series of observations of one of the sources in the above dataset. At each time, the source is selected randomly and its value at this time is added to form a new univariate series. The final series is composed of this series, the durations between random times and the indicators of the 'available source' at each time.

The details of the simulation process are presented in Appendix D. We consider synchronous and asynchronous series $X_{K \times N}$ where $K \in \{16, 64\}$ is the number of sources and $N = 10,000$, which gives 4 artificial series in total[3].

---

[3]Note that a series with $K$ sources is $K + 1$-dimensional in synchronous case and $K + 2$-dimensional in asynchronous case. The base series in all processes was a stationary AR(10) series. Although that series has the true order of 10, in the experimental setting the input data included past 60 observations. The rationale behind that is twofold: not only is the data observed in irregular random times but also in real–life problems the order of the model is unknown. Figure 6 (available in Appendix D) presents samples from the two of the simulated series.

**Electricity data**

The household electricity dataset[4] contains measurements of 7 different quantities related to electricity consumption in a single household, recorded every minute for 47 months, yielding over 2 million observations. Since we aim to focus on asynchronous time-series, we alter it so that a single observation contains only value of one of the seven features, while durations between consecutive observations range from 1 to 7 minutes[5].The regression aim is to predict all of the features at the next time step.

**Non-anonymous quotes**

The proposed model was designed primarily for forecasting incoming non-anonymous quotes received from the credit default swap market. The dataset contains 2.1 million quotes from 28 different *sources*, i.e. market participants. Each quote is characterized by 31 features: the offered price, 28 indicators of the quoting source, the *direction* indicator (the quote refers to either a buy or a sell offer) and duration from the previous quote. For each source and direction we aim at predicting the next quoted price from this given source and direction considering the last 60 quotes. We formed 15 separate prediction tasks; in each task the model was trained to predict the next quote by one of the fifteen most active market participants[6].

This dataset, which is proprietary, motivated the aforementioned construction of artificial *asynchronous* time series datasets based on its statistical features for reproducible research purpose.

## 5.2 RESULTS

Table 1 presents the detailed results from the artificial and electricity datasets. The proposed networks outperform significantly the benchmark networks on the asynchronous, electricity and quotes datasets. For the synchronous datasets, on the other hand, SOCNN almost matches the results of the benchmarks. This similar performance could have been anticipated - the correct weights of the past values in synchronous artificial datasets are far less nonlinear than in case when separate dimensions are observed asynchronously. For this reason, the significance network's potential is not fully utilized. We can also observe that the depth of the offset network has negligible or negative

Table 1: Detailed results for all datasets. For each model, we present the mean squared error obtained on the out-of-sample test set. The best results for each dataset are marked by bold font. *SOCNN1* (*SOCNN1+*) denote proposed models with one (10 or 7) offset sub-network layers. For quotes dataset the presented values are averaged mean-squared errors from 6 separate prediction tasks, normalized according to the error obtained by VAR model.

| model | VAR | CNN | ResNet | LSTM | SOCNN1 | SOCNN1+ |
|---|---|---|---|---|---|---|
| Synchronous 16 | 0.841 | 0.152 | **0.150** | 0.152 | 0.154 | 0.173 |
| Synchronous 64 | 0.364 | **0.028** | **0.028** | **0.028** | 0.029 | 0.031 |
| Asynchronous 16 | 0.577 | 0.040 | 0.032 | 0.027 | **0.017** | 0.020 |
| Asynchronous 64 | 0.318 | 0.041 | 0.046 | 0.050 | **0.032** | 0.034 |
| Electricity | 0.729 | 0.366 | 0.359 | 0.463 | **0.158** | **0.158** |
| Quotes | 1.000 | 0.891 | 2.245 | 0.841 | **0.374** | – |

impact on the results achieved by the SOCNN network. This means that the significance network has crucial impact on the performance, which is in-line with the potential drawbacks of the LSTM network discussed in Section 3: obtaining proper weights for the past observations is much more challenging than getting good predictors from the single past values.

---

[4]Available at UCI Machine Learning Repository website
https://archive.ics.uci.edu/ml/datasets/individual+household+electric+power+consumption
[5]The exact details of preprocessing can be found in Appendix E.1.
[6]This separation is related to data normalization purposes and different magnitudes of the levels of predictability for different market participants. The quotes from the remaining 13 participants were not selected for prediction as their market presence was too short or too irregular to form reliable training, validation and test samples.

Table 2: MSE for different values of $\alpha$ for two artificial datasets.

| $\alpha$ | Async 16 | async 64 |
|------|----------|----------|
| 0.0 | 0.0284 | 0.0624 |
| 0.01 | 0.0253 | 0.0434 |
| 0.1 | 0.0172 | 0.0323 |

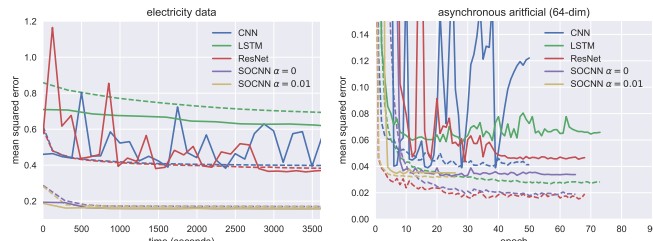

Figure 4: Learning curves for CNNs and SOCNNs with different auxiliary weights for two datasets. The solid lines indicate the test error while the dashed lines indicate the training error. Note the different scales on the horizontal axes.

For Quotes dataset, the proposed model was the best one for all 15 tasks and the only one to always beat VAR model. Surprisingly, for each of the other networks it was difficult to excel the benchmark set by simple linear model. We also found benchmark networks to have very unstable test loss during training in some cases, despite convergence of the training error. Especially, for one of the tasks LSTM and ResNet obtained very high test errors[7]. The auxiliary loss was usually found to have minor importance, though in some cases it led to best results.

The small positive auxiliary weight helped achieve more stable test error throughout training in many cases. The higher weights of auxiliary loss considerably improved the test error on asynchronous datasets (See Table 2); however for other datasets its impact was negligible. In general, the proposed SOCNN had significantly lower variance of the test and validation errors, especially in the early stage of the training process and for quotes dataset. Figure 4 presents the learning curves for two different artificial datasets.

**Model robustness**

To understand better why SOCNN obtained better results than the other networks, we check how these networks react to the presence of additional noise in the input terms[8]. Figure 5 presents changes in the mean squared error and significance and offset network outputs with respect to the level of noise. SOCNN is the most robust out of the compared networks and, together with single-layer LSTM, least prone to overfitting. Despite the use of dropout and cross-validation, the other models tend to overfit the training set and have non-symmetric error curves on test dataset.

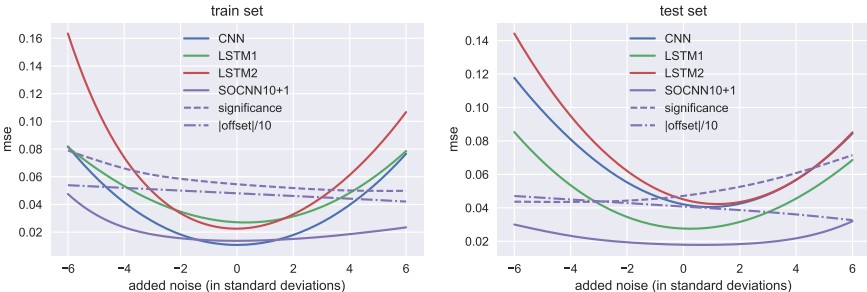

Figure 5: Experiment comparing robustness of the considered networks for *Asynchronous 16* dataset. The plots show how the error would change if an additional noise term was added to the input series. The dotted curves show the total significance and average absolute offset (not to scale) outputs for the noisy observations. Interestingly, significance of the noisy observations increases with the magnitude of noise; i.e. noisy observations are far from being discarded by SOCNN.

---

[7]The exact results for all tasks for Quotes dataset can be found in Appendix F.

[8]The details of the added noise terms are presented in the Appendix B.

# 6 CONCLUSION AND DISCUSSION

In this article, we proposed a weighting mechanism that, coupled with convolutional networks, forms a new neural network architecture for time series prediction. The proposed architecture is designed for regression tasks on asynchronous signals in the presence of high amount of noise. This approach has proved to be successful in forecasting financial and artificially generated asynchronous time series outperforming popular convolutional and recurrent networks.

The proposed model can be further extended by adding intermediate weighting layers of the same type in the network structure. Another possible generalization that requires further empirical studies can be obtained by leaving the assumption of independent offset values for each past observation, i.e. considering not only 1x1 convolutional kernels in the offset sub-network.

Finally, we aim at testing the performance of the proposed architecture on other real-life datasets with relevant characteristics. We observe that there exists a strong need for common 'econometric' datasets benchmark and, more generally, for time series (stochastic processes) regression.

ACKNOWLEDGEMENTS

Authors thank Engineering and Physical Sciences Research Council (EPSRC) for financial support for this research.

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

## APPENDIX A    NONLINEARITY IN THE ASYNCHRONOUSLY SAMPLED AUTOREGRESSIVE TIME SERIES

**Lemma 1.** *Let $X(t)$ be an AR(2) time series given by*

$$X(t) = aX(t-1) + bX(t-2) + \varepsilon(t), \tag{15}$$

*where $(\varepsilon(t))_{t=1,2,\dots}$ are i.i.d. error terms. Then*

$$\mathbb{E}[X(t)|X(t-1), X(t-k)] = a_k X(t-1) + b_k X(t-k), \tag{16}$$

*for any $t > k \geq 2$, where $a_k, b_k$ are rational functions of $a$ and $b$.*

*Proof.* The proof follows a simple induction. It is sufficient to show that

$$w_k X(t) = v_k X(t-1) + b^{k-1} X(t-k) + E_k(t), \quad k \geq 2, \tag{17}$$

where $w_k = w_k(a, b), v_k = v_k(a, b)$ are polynomials given by

$$(w_2, v_2) = (1, a) \tag{18}$$
$$(w_{k+1}, v_{k+1}) = (-v_k, -(bw_k + av_k)), \quad k \geq 2, \tag{19}$$

and $E_k(t)$ is a linear combination of $\{\varepsilon(t-i), i = 0, 1, \ldots, k-2\}$. Basis of the induction is trivially satisfied via 15. In the induction step, we assume that 17 holds for $k$. For $t > k+1$ we have $w_k X(t-1) = v_k X(t-2) + b^{k-1} X(t-k-1) + E_k(t-1)$. Multiplying sides of this equation by $b$ and adding $av_k X(t-1)$ we obtain

$$(av_k + bw_k)X(t-1) = v_k(aX(t-1) + bX(t-2)) + b^k X(t-k-1) + bE(t-1). \tag{20}$$

Since $aX(t-1) + bX(t-2) = X(t) - \varepsilon(t)$ we get

$$-v_{k+1} X(t-1) = -w_{k+1} X(t) + b^k X(t-k-1) + [bE_k(t-1) - v_k \varepsilon(t)] \tag{21}$$

As $E_{k+1}(t) = bE_k(t-1) - v_k \varepsilon(t)$ is a linear combination of $\{\varepsilon(t-i), i = 0, 1, \ldots, k-1\}$, the above equation proves 17 for $k = k+1$.

$\square$

## APPENDIX B    ROBUSTNESS OF THE PROPOSED ARCHITECTURE

To see how robust each of the networks is, we add noise terms to part of the input series and evaluate them on such datapoints, assuming unchanged output. We consider varying magnitude of the noise terms, which are added only to the selected 20% of past steps at the value dimension[9]. Formally the procedure is following:

1. Select randomly $N_{obs} = 6000$ observations $(X_n, y_n)$ (half of which coming from training set and half from test set).

2. Add noise terms to the observations $\widetilde{X}_n^p := X_n + \Xi_n \cdot \gamma_p$, for $\{\gamma_p\}_{p=1}^{128}$ evenly distributed on $[-6\sigma, 6\sigma]$, where $\sigma$ is a standard deviation of the differences of the series being predicted and

$$(\Xi_n)_{tj} = \begin{cases} \xi_n \sim \mathcal{U}[0, 1] & \text{if } j = 0, t \in [0, 5, \ldots, 55] \\ 0 & \text{otherwise.} \end{cases} \tag{22}$$

3. For each $p$ evaluate each of the trained models on dataset $\left\{\widetilde{X}_n^p, y_n\right\}_{n=1}^{N_{obs}}$, separately for $n$'s originally coming from training and test sets.

## APPENDIX C    TRAINING DETAILS

### C.1    NETWORK SETTINGS

To evaluate the model and the significance of its components, we perform a grid search over some of the hyperparameters, more extensively on the artificial and electricity datasets. These include the offset sub-network's depth (we consider depths of 1 and 10 for artificial and electricity datasets; 1 for Quotes data) and the auxiliary weight $\alpha$ (compared values: $\{0, 0.1, 0.01\}$). For all networks we have chosen LeakyReLU activation function (23)

$$\sigma^{LeakyReLU}(x) = x \text{ if } x \geq 0, \quad ax \text{ otherwise.} \tag{23}$$

with leak rate $a = .1$ as an activation function.

---

[9]The asynchronous series has one dimension representing the *value* of the quote, one representing *duration* and others representing indicators of the *source*. See **??** for details.

## C.2 Benchmark networks

We compare the performance of the proposed model with CNN, ResNet, multi-layer LSTM networks and linear (VAR) model. The benchmark networks were designed so that they have a comparable number of parameters as the proposed model. Consequently, LeakyReLU activation function (23) with leak rate .1 was used in all layers except the top ones where linear activation was applied. For CNN we provided the same number of layers, same stride (1) and similar kernel size structure. In each trained CNN, we applied max pooling with the pool size of 2 every two convolutional layers[10]. Table 3 presents the configurations of the network hyperparameters used in comparison.

Table 3: Configurations of the trained models. $f$ - number of convolutional filters/memory cell size in LSTM, $ks$ - kernel size, $p$ - dropout rate, $clip$ - gradient clipping threshold, $conv$ - $(k \times 1)$ convolution with kernel size $k$ indicated in the ks column, $conv1$ - $(1 \times 1)$ convolution. Apart from the listed layers, each network has a single fully connected layer on the top. Kernel sizes (3, 1) ((1, 3, 1)) denote alternating kernel sizes 3 and 1 (1, 3 and 1) in successive convolutional layers

| **Artificial & Electricity Datasets** | | | | | |
|---|---|---|---|---|---|
| Model | layers | f | ks | p | clip |
| SOCNN | 10conv + $\{1, 10\}$conv1 | $\{8, 16\}$ | $\{(3, 1), 3\}$ | 0 | $\{1, .001\}$ |
| CNN | 7conv + 3pool | $\{16, 32\}$ | $\{(3, 1), 3\}$ | $\{0, .5\}$ | $\{1, .001\}$ |
| LSTM | $\{1, 2, 3, 4\}$ | $\{16, 32\}$ | - | $\{0, .5\}$ | $\{1, .001\}$ |
| ResNet | 22conv + 3pool | 16 | (1, 3, 1) | $\{0, .5\}$ | $\{1, .001\}$ |
| **Quotes Dataset** | | | | | |
| Model | layers | f | ks | p | clip |
| SOCNN | 7conv + $\{1, 7\}$conv1 | 8 | $\{(3, 1), 3\}$ | .5 | .01 |
| CNN | 7conv + 3pool | $\{16, 32\}$ | $\{(3, 1), 3\}$ | .5 | .01 |
| LSTM | $\{1, 2, 3\}$ | $\{32\}$ | - | .5 | .0001[11] |
| ResNet | 22conv + 3pool | 16 | (1, 3, 1) | .5 | .01 |

## C.3 Network Training

The training and validation sets were sampled randomly from the first 80% of timesteps in each series, with ratio 3 to 1. The remaining 20% of data was used as a test set. All models were trained using Adam optimizer (Kingma & Ba, 2015) which we found much faster than standard Stochastic Gradient Descent in early tests. We used batch size of 128 for artificial data and 256 for quotes dataset. We also applied batch normalization (Ioffe & Szegedy, 2015) in between each convolution and the following activation. At the beginning of each epoch, the training samples were shuffled. To prevent overfitting we applied *dropout* and *early stopping*[12]. Weights were initialized following the *normalized uniform* procedure proposed by Glorot & Bengio (2010). Experiments were carried out using implementation relying on Tensorflow (Abadi et al., 2016) and Keras front end (Chollet, 2015). For artificial data we optimized the models using one K20s NVIDIA GPU while for quotes dataset we used 8-core Intel Core i7-6700 CPU machine only.

## Appendix D   Artificial data generation

We simulate a multivariate time series composed of $K$ noisy observations of the same autoregressive signal. The simulated series are constructed as follows:

---

[10]Hence layers $3, 6$ and $9$ were pooling layers, while layers $1, 2, 4, 5, \ldots$ were convolutional layers.

[11]We found LSTMs very unstable on quotes dataset without gradient clipping or with higher clipping boundary.

[12]Whenever 10 consecutive epochs did not bring improvement in the validation error, the learning rate was reduced by a factor of 10 and the best weights obtained till then were restored. After the second reduction and another 10 consecutive epochs without improvement, the training was stopped. The initial learning rate was set to .001.

1. We simulate univariate stationary AR(10) time series $x$ with randomly chosen weights.

2. The series is copied $K$ times and each copy $x^{(k)}$ is associated with a separate noise process $\varepsilon^{(k)}$. We consider Gaussian or Binomial noise of different scales; for each copy it is either added to or multiplied by the initial series ($x^{(k)} = x + \varepsilon^{(k)}$ or $x^{(k)} = x \times \varepsilon^{(k)}$).

3. We simulate a random time process $T$ where differences between consecutive events are iid exponential random variables.

4a. The final series is composed of $K$ noisy copies of the original process observed at times indicated by the random time process, and a duration between observations.

4b. At each time $T(t)$ indicated by the random time process $T$, one of the noisy copies $k$ is drawn and its value at this time $x_{T(t)}^{(k)}$ is selected to form a new noisy series $x^*$. The final multivariate series is composed of $x^*$, the series of durations between observations and $K$ indicators of which observation was drawn at each time.

Assume that $(x_t)_{t=1,2,\ldots}$ is a stationary $AR(\nu)$ series and consider the following (random) noise functions

$$
\begin{aligned}
\varepsilon_0(x, c, p) &= x + c(2\epsilon - 1), & \epsilon &\sim \text{Bernoulli}(p), \\
\varepsilon_1(x, c, p) &= x(1 + c(2\epsilon - 1)), & \epsilon &\sim \text{Bernoulli}(p), \\
\varepsilon_2(x, c, p) &= x + c\epsilon, & \epsilon &\sim \mathcal{N}(0, 1), \\
\varepsilon_3(x, c, p) &= x(1 + c\epsilon), & \epsilon &\sim \mathcal{N}(0, 1).
\end{aligned}
\tag{24}
$$

Note that argument $p$ of $\varepsilon_2$ and $\varepsilon_3$ is redundant and was added just for notational convenience.

Let $N_t \sim \text{Exp}(\lambda)$ be a series of i.i.d. exponential random variables with some constant rate $\lambda$ and let $T(t) = \sum_{s=1}^{t} \lceil N_s + 1 \rceil$. Then $T(t)$ is a strictly increasing series of times, at which we will observe the noisy observations.

Let $p_1, p_2, \ldots, p_K \in (0, 1)$ and define

$$
X_t^{(k)} := \begin{cases} \varepsilon_{k \pmod 4}(x_{T(t)}, 2^{-\lfloor k/8 \rfloor}, p_k), & k = 1, \ldots, K, \\ T(t), & k = K + 1. \end{cases}
\tag{25}
$$

Let $I(t)$ be a series of i.i.d. random variables taking values in $\{1, 2, \ldots, K\}$ such that $\mathbb{P}(I(t) = $

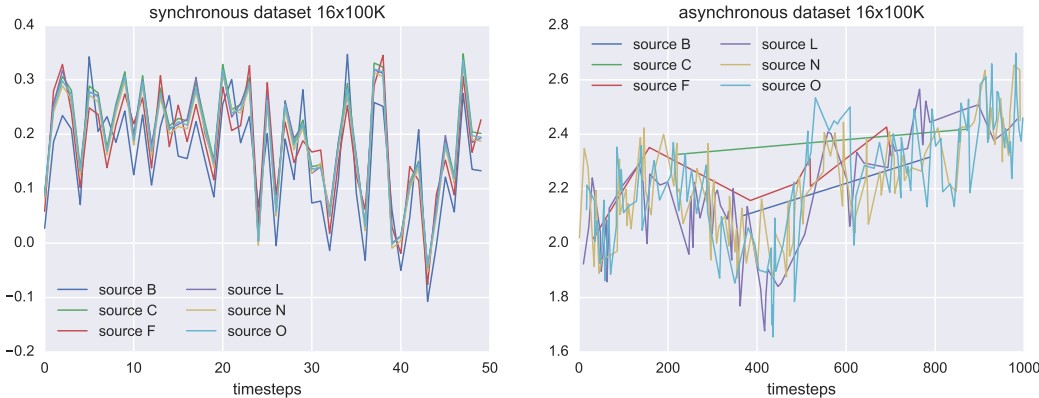

Figure 6: Simulated synchronous (left) and asynchronous (right) artificial series. Note the different durations between the observations from different sources in the latter plot. For clarity, we present only 6 out of 16 total dimensions.

$K) \propto q^K$ for some $q > 0$. Define

$$
\bar{X}_t^{(k)} := \begin{cases} 1, & k \leq K \text{ and } k = I(t), \\ 0, & k \leq K \text{ and } k \neq I(t), \\ X_t^{(I(t))}, & k = K + 1, \\ T(t), & k = K + 2. \end{cases}
\tag{26}
$$

We call $\{X_t\}_{t=1}^N$ and $\{\bar{X}_t\}_{t=1}^N$ *synchronous* and *asynchronous* time series, respectively. We simulate both of the processes for $N = 10,000$ and each $K \in \{16, 64\}$.

## APPENDIX E  HOUSEHOLD ELECTRICITY DATASET

### E.1  SAMPLING

The original dataset has 7 features: global active power, global reactive power, voltage, global intensity, sub-metering 1, sub-metering 2 and sub-metering 3, as well as information on date and time. We created asynchronous version of this dataset in two steps:

1. Deterministic time step sampling. The durations between the consecutive observations are periodic and follow a scheme [1min, 2min, 3min, 7min, 2min, 2min, 4min, 1min, 2min, 1min]; the original observations in between are discarded. In other words, if the original observations are indexed according to time (in minutes) elapsed since the first observation, we keep the observations at indices $n$ such that $n \mod 25 \equiv k \in [0, 1, 3, 6, 13, 15, 17, 21, 22, 24]$.

2. Random feature sampling. At each remaining time step, we choose one out of seven features that will be available at this step. The probabilities of the features were chosen to be proportional to $[1, 1.5, 1.5^2, 1.5^6]$ and randomly assigned to each feature before sampling (so that each feature has constant probability of its value being available at each time step.

At each time step the sub-sampled dataset is 10-dimensional vector that consists information about the time, date, 7 indicator features that imply which feature is available, and the value of this feature. The length of the sub-sampled dataset is above 800 thousand, i.e. 40% of the original dataset's length.

The schedule of the sampled timesteps and available features is attached in the *data* folder in the supplementary material.

### E.2  SIGNIFICANCE AND OFFSET ACTIVATIONS

In Figure 7 we present significance and offset activations for three input series, from the network trained on *electricity* dataset. Each row represents activations corresponding to past values of a single feature.

## APPENDIX F  DETAILED RESULTS FOR QUOTES DATASET

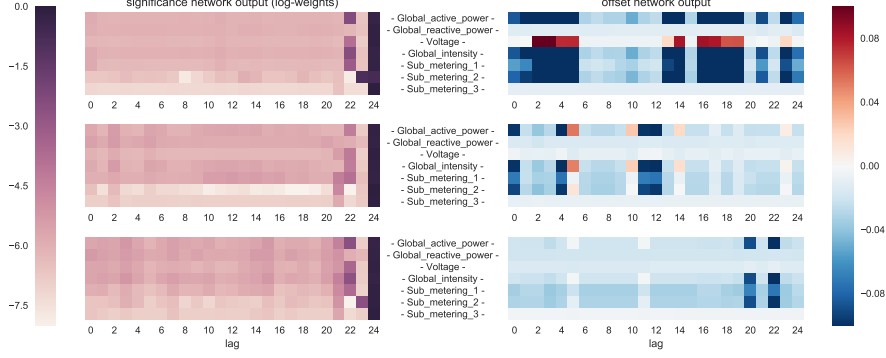

Figure 7: Activations of the *significance* and *offset* sub-networks for the network trained on Electricity dataset. We present 25 most recent out of 60 past values included in the input data, for 3 separate datapoints. Note the log scale on the left graph.

Table 4: Detailed results for each prediction task for the quotes dataset. Each task involves prediction of the next quote by one of the banks. Numbers represent the mean squared errors on out-of-sample test set.

| task | CNN | VAR | LSTM | ResNet | SOCNN |
|---|---|---|---|---|---|
| bank A | 0.993 | 1.123 | 0.999 | 1.086 | **0.530** |
| bank B | 1.225 | 2.116 | 1.673 | 31.598 | **0.613** |
| bank C | 3.208 | 3.952 | 2.957 | 3.805 | **0.617** |
| bank D | 3.634 | 4.134 | 3.436 | 4.635 | **0.649** |
| bank E | 3.558 | 4.367 | 3.344 | 3.717 | **1.154** |
| bank F | 8.541 | 8.150 | 7.880 | 8.274 | **1.553** |
| bank G | 0.248 | 0.278 | 0.132 | 1.462 | **0.063** |
| bank I | 4.777 | 4.853 | 3.933 | 4.936 | **0.400** |
| bank J | 1.094 | 1.172 | 1.097 | 1.216 | **0.773** |
| bank K | 2.521 | 4.307 | 2.573 | 4.731 | **0.926** |
| bank L | 1.108 | 1.448 | 1.186 | 1.312 | **0.743** |
| bank M | 1.743 | 1.816 | 1.741 | 1.808 | **1.271** |
| bank N | 3.058 | 3.232 | 2.943 | 3.229 | **1.509** |
| bank O | 0.539 | 0.532 | 0.420 | 0.566 | **0.218** |
| bank P | 0.447 | 0.354 | 0.470 | 0.510 | **0.283** |

