# OpenReview forum: "Autoregressive Convolutional Neural Networks for Asynchronous Time Series"
_ICLR.cc/2018/Conference — Reject_

### Official Review · AnonReviewer2 · 2017-11-18
**Missing lots of related work**

**Rating:** 4
**Confidence:** 5

**Review:**

To begin with, the authors seem to be missing some recent developments in the field of deep learning which are closely related to the proposed approach; e.g.:

Sotirios P. Chatzis, “Recurrent Latent Variable Conditional Heteroscedasticity,” Proc. 42nd IEEE International Conference on Acoustics, Speech and Signal Processing (IEEE ICASSP), pp. 2711-2715, March 2017.

In addition, the authors claim that Gaussian process-based models are not appropriate for handling asynchronous data, since the assumed Gaussianity is inappropriate for financial datasets, which often follow fat-tailed distributions. However, they seem to be unaware of several developments in the field, where mixtures of Gaussian processes are postulated, so as to allow for capturing long tails in the data distribution; for instance:

Emmanouil A. Platanios and Sotirios P. Chatzis, “Gaussian Process-Mixture Conditional Heteroscedasticity,” IEEE Transactions on Pattern Analysis and Machine Intelligence, vol. 36, no. 5, pp. 888-900, May 2014.

Hence, the provided experimental comparisons are essentially performed against non-rivals of the method. It is more than easy to understand that a method not designed for modeling observations with the specific characteristics of financial data should definitely perform worse than a method designed to cope with such artifacts. That is why the sole purpose of a (convincing) experimental evaluation regime should be to compare between methods that are designed with the same data properties in mind. The paper does not satisfy this requirement.

Turning to the method itself, the derivations are clear and straightforward; the method could have been motivated a somewhat better, though.

---

### Official Review · AnonReviewer3 · 2017-11-18
**There are some novel ideas in this paper, but it's not entirely clear if the ideas are all that useful. The experiments are okay, but don't really highlight the usefulness of the individual proposed ideas very much.**

**Rating:** 5
**Confidence:** 3

**Review:**

The author proposed:
1. A data augmentation technique for asynchronous time series data.
2. A convolutional 'Significance' weighting neural network that assigns normalised weights to the outputs of a fully-connected autoregressive 'Offset' neural network, such that the output is a weighted average of the 'Offset' neural net.
3. An 'auxiliary' loss function.

The experiments showed that:
1. The proposed method beat VAR/CNN/ResNet/LSTM 2 synthetic asynchronous data sets, 1 real electricity meter data set and 1 real financial bid/ask data set. It's not immediately clear how hyper-parameters for the benchmark models were chosen.
2. The author observed from the experiments that the depth of the offset network has negligible effect, and concluded that the 'Significance' network has crucial impact. (I don't see how this conclusion can be made.)
3. The proposed auxiliary loss is not useful.
4. The proposed architecture is more robust to noise in the synthetic data set compared to the benchmarks, and together with LSTM, are least prone to overfitting.

Pros
- Proposed a useful way of augmenting asynchronous multivariate time series for fitting autoregressive models
- The convolutional Significance/weighting networks appears to reduce test errors (not entirely clear)

Cons
- The novelties aren't very well-justified. The 'Significance' network was described as critical to the performance, but there is no experimental result to show the sensitivity of the model's performance with respect to the architecture of the 'Significance' network. At the very least, I'd like to see what happens if the weighting was forced to be uniform while keeping the 'Offset' network and loss unchanged.
- It's entirely unclear how the train and test data was split. This may be quite important in the case of the financial data set.
- It's also unclear if model training was done on a rolling basis, which is common for time series forecasting.
- The auxiliary loss function does not appear to be very helpful, but was described as a key component in the paper.

Quality: The quality of the paper was okay. More details of the experiments should be included in the main text to help interpret the significance of the experimental results. The experiment also did not really probe the significance of the 'Significance' network even though it's claimed to be important.
Clarity: Above average.
Originality: Mediocre. Nothing really shines. Weighted average-type architecture has been proposed many times in neural networks (e.g., attention mechanisms).
Significance: Low. It's unclear how useful the architecture really is.

---

### Official Review · AnonReviewer1 · 2017-11-27
**The paper has potential but needs more empirical validation to demonstrate general relevance**

**Rating:** 5
**Confidence:** 4

**Review:**

The authors propose an extension to CNN using an autoregressive weighting for asynchronous time series applications.  The method is applied to a proprietary dataset as well as a couple UCI problems and a synthetic dataset, showing improved performance over baselines in the asynchronous setting.

This paper is mostly an applications paper.  The method itself seems like a fairly simple extension for a particular application, although perhaps the authors have not clearly highlighted details of methodological innovation.  I liked that the method was motivated to solve a real problem, and that it does seem to do so well compared to reasonable baselines.  However, as an an applications paper, the bread of experiments are a little bit lacking -- with only that one potentially interesting dataset, which happens to proprietary.  Given the fairly empirical nature of the paper in general, it feels like a strong argument should be made, which includes experiments, that this work will be generally significant and impactful.

The writing of the paper is a bit loose with comments like:
“Besides these and claims of secretive hedge funds (it can be marketing surfing on the deep learning hype), no promising results or innovative architectures were publicly published so far, to the best of our knowledge.”

Parts of the also appear rush written, with some sentences half finished:
“"ues of x might be heterogenous, hence On the other hand, significance network provides data-dependent weights for all regressors and sums them up in autoregressive manner.””

As a minor comment, the statement
“however, due to assumed Gaussianity they are inappropriate for financial datasets, which often follow fat-tailed distributions (Cont, 2001).”
Is a bit too broad.  It depends where the Gaussianity appears.  If the likelihood is non-Gaussian, then it often doesn’t matter if there are latent Gaussian variables.

---

### Author Response · Authors · 2018-01-05
**Response to the reviewers**

First of all, we would like to thank all reviewers for their insightful comments.

As the ratings quite consistently put the paper below the acceptance threshold, the authors decided not to modify the submission, but instead to continue the work and possibly re-submit the paper in the future. We agree that the obtained results lack comparison with other significant models as well as the proposed model without certain components (e.g. significance network).  New experiments will be carried out to reinforce the results.

---

### Decision · Program_Chairs · 2018-01-29
**ICLR 2018 Conference Acceptance Decision**

**Decision:**

Reject

**Comment:**

The reviewers feel that the novelties in the model are not significant.   Furthermore, they suggest that empirical results could be improved by
1:  analyses showing how the significance network functions and directly measuring its impact
2: More reproducible experiments.  In particular, this is really an applications paper, and the experiments on the main application are not reproducible because the data is proprietary.
3: baselines that make assumptions more in line with the authors' problem setup